# Metabolomic Investigation of Blood and Urinary Amino Acids and Derivatives in Patients with Type 2 Diabetes Mellitus and Early Diabetic Kidney Disease

**DOI:** 10.3390/biomedicines11061527

**Published:** 2023-05-25

**Authors:** Maria Mogos, Carmen Socaciu, Andreea Iulia Socaciu, Adrian Vlad, Florica Gadalean, Flaviu Bob, Oana Milas, Octavian Marius Cretu, Anca Suteanu-Simulescu, Mihaela Glavan, Silvia Ienciu, Lavinia Balint, Dragos Catalin Jianu, Ligia Petrica

**Affiliations:** 1Department of Internal Medicine II–Division of Nephrology, “Victor Babes” University of Medicine and Pharmacy Timisoara, County Emergency Hospital Timisoara, Eftimie Murgu Sq. No. 2, 300041 Timisoara, Romania; maria.stefan2014@yahoo.com (M.M.); flaviu_bob@yahoo.com (F.B.); oana.milas@yahoo.com (O.M.); anca.simulescu@yahoo.com (A.S.-S.); mihaelapatruica@gmail.com (M.G.); ienciu.silviaoana@yahoo.com (S.I.); lavinia.balint@umft.ro (L.B.); ligia_petrica@yahoo.co.uk (L.P.); 2Centre for Molecular Research in Nephrology and Vascular Disease, Faculty of Medicine, “Victor Babes” University of Medicine and Pharmacy, Eftimie Murgu Sq. No. 2, 300041 Timisoara, Romania; csocaciudac@gmail.com (C.S.); vlad.adrian@umft.ro (A.V.); jianu.dragos@umft.ro (D.C.J.); 3Research Center for Applied Biotechnology and Molecular Therapy BIODIATECH, SC Proplanta, Str. Trifoiului 12G, 400478 Cluj-Napoca, Romania; 4Department of Occupational Health, University of Medicine and Pharmacy “Iuliu Haţieganu”, Str. Victor Babes 8, 400347 Cluj-Napoca, Romania; andreeaiso@gmail.com; 5Department of Internal Medicine II–Division of Diabetes and Metabolic Diseases, “Victor Babes” University of Medicine and Pharmacy Timisoara, County Emergency Hospital Timisoara, Eftimie Murgu Sq. No. 2, 300041 Timisoara, Romania; 6Department of Surgery I–Division of Surgical Semiology I, “Victor Babes” University of Medicine and Pharmacy Timisoara, Emergency Clinical Municipal Hospital Timisoara, Eftimie Murgu Sq. No. 2, 300041 Timisoara, Romania; tavicretu@yahoo.com; 7Department of Neurosciences–Division of Neurology, “Victor Babes” University of Medicine and Pharmacy Timisoara, County Emergency Hospital Timisoara, Eftimie Murgu Sq. No. 2, 300041 Timisoara, Romania; 8Centre for Cognitive Research in Neuropsychiatric Pathology (Neuropsy-Cog), Faculty of Medicine, “Victor Babes” University of Medicine and Pharmacy, Timisoara, Eftimie Murgu Sq. No. 2, 300041 Timisoara, Romania; 9Center for Translational Research and Systems Medicine, Faculty of Medicine, “Victor Babes” University of Medicine and Pharmacy, Eftimie, Murgu Sq. No. 2, 300041 Timisoara, Romania

**Keywords:** metabolomics, diabetic kidney disease, HPLC-MS, amino acids

## Abstract

Diabetic kidney disease (DKD) is the leading cause of end-stage renal disease; however, few biomarkers of its early identification are available. The aim of the study was to assess new biomarkers in the early stages of DKD in type 2 diabetes mellitus (DM) patients. This cross-sectional pilot study performed an integrated metabolomic profiling of blood and urine in 90 patients with type 2 DM, classified into three subgroups according to albuminuria stage from P1 to P3 (30 normo-, 30 micro-, and 30 macroalbuminuric) and 20 healthy controls using high-performance liquid chromatography and mass spectrometry (UPLC-QTOF-ESI* MS). From a large cohort of separated and identified molecules, 33 and 39 amino acids and derivatives from serum and urine, respectively, were selected for statistical analysis using Metaboanalyst 5.0. online software. The multivariate and univariate algorithms confirmed the relevance of some amino acids and derivatives as biomarkers that are responsible for the discrimination between healthy controls and DKD patients. Serum molecules such as tiglylglycine, methoxytryptophan, serotonin sulfate, 5-hydroxy lysine, taurine, kynurenic acid, and tyrosine were found to be more significant in the discrimination between group C and subgroups P1–P2–P3. In urine, o-phosphothreonine, aspartic acid, 5-hydroxy lysine, uric acid, methoxytryptophan, were among the most relevant metabolites in the discrimination between group C and DKD group, as well between subgroups P1–P2–P3. The identification of these potential biomarkers may indicate their involvement in the early DKD and 2DM progression, reflecting kidney injury at specific sites along the nephron, even in the early stages of DKD.

## 1. Introduction

To date, diabetes mellitus has experienced a progressive increase in the number of cases worldwide, and over 40% of them will end up in the renal function replacement program through hemodialysis [1]. An early diagnosis of diabetic kidney disease (DKD) involves finding some biomarkers besides eGFR and levels of urinary albumin and creatinine ratio [2].

In recent decades, it was necessary to investigate and validate new biomarkers of proximal tubule dysfunction, oxidative stress, and inflammation [3,4].

Metabolomics consists of the identification of low molecular weight molecules which represent the intermediary and end products of cellular functions in a biological sample by using different profiling techniques such as nuclear magnetic resonance and mass spectrometry. The number of metabolites in an organism is defined as a metabolome, and the changes in a human metabolome might be useful information in the discovery of the disease [3].

Patients with DKD show changes in energy metabolism. Renal tissue is damaged by metabolic disorders such as hyperglycemia and dyslipidemia [5]. Glucose and tricarboxylic acid (TCA) cycle metabolites are deposited in diabetic renal tissue, which might be associated with mitochondrial dysfunction [6,7], with its activation being related to glucose assimilation in the hyperglycemic state [8]. This process involves increased oxygen consumption and renal hypoxia [9], which produces a large amount of reactive oxygen species (ROS), in association with mitochondrial fragmentation [10]. Consequently, the morphologic modification of mitochondria and energy metabolism changes are linked, and the effect is expressed by an excess of ROS in diabetic renal tissue.

The kidneys are also exposed to pathogenic endoplasmic stress (ER) under oxidative stress, glycate stress, and hypoxia [11]. A disturbance of unfolded protein response (UPR) pathways usually occurs in glomerular and tubulointerstitial cells. The podocyte is damaged by pathogenic ER stress [12], which is linked to the progression of glomerulonephritis [13].

Previous studies have shown robust changes in the metabolites of the tricarboxylic acid cycle, lipid metabolism, amino acid metabolism, urea cycle, and nucleotide metabolism which were associated with DKD [14]. The involvement of amino acids and their metabolites in the diagnosis and progression if DKD is still of high importance, as demonstrated by many recent publications which applied the metabolomic approach. Using untargeted metabolomics for a large cohort of DKD patients and healthy controls, at stages 1–5, five metabolites were previously identified, including 5-methoxytryptophan (5-MTP), whose levels strongly correlate with clinical markers of kidney disease [15].

Lower tryptophan levels and higher kynurenine/tryptophan ratios were significantly associated with macroalbuminuria, which may predict angiotensin receptor blocker responsiveness in DKD patients [16]. A particular interest was given to taurine, which can be a metabolic marker, since it provides protection against renal alteration (hypertension and proteinuria), specific glomerular and tubular disturbance, acute and chronic renal conditions, and diabetic nephropathy. Taurine seems to have ameliorative effects against renal disorders due to its relationship with the renin–angiotensin–aldosterone system, osmoregulatory properties, and signaling pathways [17,18]. Hippuric acid is known as a gut microbial co-metabolite of benzoic acid that is subsequently conjugated with glycine in the mitochondria and excreted in the urine. Several studies discovered that patients with diabetes had lower levels of hippuric acid than those without diabetes [19]; moreover, they had low levels in both human diabetic renal pathology studies and animal DKD models [20]. For this reason, it has been recommended as an additional indicator of DKD.

Higher symmetric (SDMA) and asymmetric dimethylarginine (ADMA) were associated with and remarkably enhanced the risk prediction for all-cause mortality in renal function beyond traditional risk factors [21].

The purpose of this study was to focus on amino acid metabolism and assess new urinary and blood serum biomarkers in the early DKD of type 2 diabetes mellitus (DM) patients. The UPLC-QTOF-ESI* MS results aimed to identify the most relevant amino acids and derivatives, as metabolic biomarkers, with a predictive value for DKD progression.

## 2. Materials and Methods

### 2.1. Patients and Compliance with Ethical Standards

Before beginning, the protocol of the study was authorized by the Ethics Committee for Scientific Research of “Victor Babes” University of Medicine and Pharmacy Timisoara (no. 28/02 September 2020) and “Pius Brinzeu” County Emergency Hospital Timisoara (no. 296/06 April 2022), including the details about samples and written consent. Ninety diabetic patients with a DM duration longer than 5 years were recruited from the Department of Nephrology and the Department of Diabetes and Metabolic Diseases and “Pius Brinzeu” County Emergency Hospital Timisoara, and 20 healthy control subjects were included in a cross-sectional study. During blood and urine collection, all patients were prescribed treatment with angiotensin-converting enzyme inhibitors/angiotensin 2 receptor blockers. Exclusion criteria included poor control of diabetes (HbA1c over 10%), cancer pathology, glomerulonephritis, active infections, and T1DM. Serum and urine were collected from 90 patients diagnosed with DKD (group P), staged using the urinary albumin/creatinine ratio (UACR) (normo- < 30 mg/g subgroup P1, micro-30-300 mg/g subgroup P2, and macroalbuminuria- > 300 mg/g subgroup P3, respectively) and 20 samples from healthy controls (group C) (Table 1).

### 2.2. Blood and Urine Collection and Processing

Sample preparation was performed following a standard protocol, as described previously [22,23].

### 2.3. UHPLC-QTOF-ESI^+^-MS Analysis

Ultra-high-performance liquid chromatography coupled with electrospray ionization-quadrupole-time of flight-mass spectrometry (UHPLC-QTOF-ESI^+^-MS) was utilized to perform the metabolite profiling. An Acclaim C18 column with a pore size of 30 nm separated the metabolites. The mobile phase consisted of 0.1% formic acid in water (A) and 0.1% formic acid in acetonitrile (B). The elution time was set for 20 min. The flow rate was set at 0.3 mL min^−1^ for serum samples and 0.8 mL min^−1^ for urine samples. Several QC samples obtained from each group were used in parallel to calibrate the separations. Doxorubicin hydrochloride was added in parallel to the QC samples as an internal standard.

The applied MS parameters included ionization mode positive (ESI^+^), MS calibration with Natrium formate, and a capillary voltage of 3500 V; the pressure for the nebulizing gas was 2.8 barr, drying gas flow was 12 L/min, and the drying temperature was 300 °C. The *m*/*z* values to be separated were set between 60 and 600 Daltons. The control of the instrument and the data processing were performed using TofControl 3.2, HyStar 3.2, Data Analysis 4.2, and Chromeleon™ 6.8 Chromatography Data System (CDS) Software, respectively.

### 2.4. Statistical Analysis

The acquired data were processed by means of the Bruker software Data Analysis 4.2, installed in the instrument. Details of the separated molecules were obtained by using a peak dissect algorithm; afterwards, a first advanced bucket matrix was generated using the Find Molecular Features (FMF) algorithm, which accounted for each *m*/*z* value, the retention time, the peak area, the peak intensity, and the signal/noise (S/N) ratio. The Total Ion Chromatograms (TICs) and base peak chromatograms (BPCs) were obtained from the total ion chromatogram using specific algorithms. The number of separated molecules (*m*/*z* values) ranged between 320 and 420 in serum samples and up to 550 in urine samples. The second step consisted of the alignment of the common molecules (with the same *m*/*z* value) in all samples using the online software from www.bioinformatica.isa.cnr.it/NEAPOLIS (accessed on 13 September 2022), maintaining the final matrix of the molecules that were common in more than 80% of the samples. Therefore, in the final matrices, the number of common molecules (*m*/*z* values) obtained from serum and urine was 136 and 196, respectively.

Since we wanted to target the amino acid metabolites and derivatives, the final number of common molecules in the blood serum and urine submitted for statistical analysis was 38. The Excel matrix (.xlsx) was converted to a .csv file, which was introduced to the Metaboanalyst 5.0 platform for multivariate and univariate analysis (https://www.metaboanalyst.ca/MetaboAnalyst/ModuleView.xhtml, accessed on 13 September 2022). Statistical analysis was performed using Metaboanalyst 5.0 software, as well as partial least squares discriminant analysis (PLSDA), variable importance in projection (VIP) scores, Random Forest scores, and biomarker analysis (ROC curves), to identify potential biomarkers that differentiate the DKD group and the control subjects.

## 3. Results

The untargeted metabolomic analysis was performed using multivariate and univariate analysis and by comparing control group C (healthy subjects) with the pathologic group P, including subgroups (P1 to P3) ranked according to their increase in urinary albumin/creatinine ratio and decrease in eGFR.

The results were presented graphically, and the significant biomarkers of differentiation were identified using Metaboanalyst 5.0 software. The identification of molecules, based on their *m*/*z* value and the retention time, was performed in agreement with our database and other international databases for metabolomics, such as the Human Metabolome Database (http://www.hmdb.ca), Lipid Maps (http://www.lipidmaps.org), and PubChem (https://pubchem.ncbi.nlm.nih.gov).

### 3.1. Multivariate Analysis of Blood Serum

#### 3.1.1. PLSDA and Volcano Plot

Significant discrimination was observed between groups C (code 0) and P (code 1), with a co-variance of 46%. The cross-validation algorithm based on PLSDA analysis showed the highest accuracy, with high R2 values (>0.95) and a significant Q2 value (>0.93) for the third component, confirming the good predictability of this model. According to the volcano plot and log2(FC) values (positive and negative, which means decreases in levels in group P vs. C, or increases, respectively), the names of the molecules which contributed to a difference between the C and P groups are shown. Table 1 describes the FC values and the log2(FC) combined with the VIP values from the PLSDA analysis (Figure 1).

#### 3.1.2. Fold Change Values, VIP Scores, and Random Forest Data

The FC and log2(FC) values, according to the volcano plot and the VIP scores from PLSDA analysis ranging from 1.5 to 0.950, were considered relevant and the respective molecules were selected (Table 2). Moreover, in the same table, the Mean Decrease Accuracy (MDA) values obtained using Random Forest analysis are presented (Table 2).

#### 3.1.3. Heatmap

The heatmap plot (Figure 2) illustrates the different clustering of the subgroups 0 and 1 as well as the relationships between molecules (increase or decrease in the groups 0 and 1).

#### 3.1.4. Biomarker Analysis

Based on the results of the biomarker analysis, the highest AUC values for the molecules to be considered the best biomarkers of the differentiation between groups P and C are presented in Table 3.

### 3.2. Univariate Analysis (One-Way ANOVA) to Evaluate the Relationship between Blood Metabolites and DKD/2DM Progression

#### 3.2.1. ANOVA Parameters and Fisher’s LSD, VIP Scores, and MDA Values

Table 4 represents the f-values and *p*-values as well the Fisher’s LSD significant correlations between the controls (group C) and subgroups P1–P3, which are classified according to increased UACR and decreased eGFR (see Table 2).

#### 3.2.2. PLSDA and Heatmap

Significant discrimination was observed between group C and subgroups P1–P3, with a co-variance of 47.4%. Meanwhile, acceptable discrimination between subgroups P1–P2–P3 was observed, considering component 2. The cross-validation algorithm presented a good accuracy in this case (0.76), with high R2 values (0.82) and a significant Q2 value (0.75) for the fourth component, confirming the good predictability of the model. The heatmap illustrates the good classification of sample groups, as well as the clusters of group C vs. subgroups P1–P3 and their relative variation (increase or decrease). There is a clear delimitation between the molecules with lower levels in group C compared to subgroups P1–P3, as can be seen in the heatmap (from tiglylglycine to spermidine). Moreover, compared to molecules converted from aspartic acid to acetyl serine, there are higher levels in the group C (Figure 3).

### 3.3. Multivariate Analysis of Urine

#### 3.3.1. PLSDA and Volcano Plot

The cross-validation algorithm presented a maximal accuracy with high R2 values (0.9) and a significant Q2 value (0.8) for the second component, confirming the good predictability of the model. According to the volcano plot and log2(FC) values (positive and negative, which suggests a decrease in the levels of group P vs. C, or increases, respectively), the name of the molecules which made the difference between groups C and P are shown. Table 5 describes the FC values and the log2(FC) combined with VIP values from PLSDA analysis (Figure 4).

#### 3.3.2. Fold Change Values, VIP Scores, and Random Forest Data

The FC and log2(FC) values and the VIP scores from the PLSDA analysis ranging from 1.906 to 0.90 were considered relevant, and the respective molecules were selected (Table 5). Moreover, in the same table, according to the results of the random forest analysis, the Mean Decrease Accuracy (MDA) values are presented.

#### 3.3.3. Heatmap

The heatmap plot (Figure 5) illustrates the different clustering of the subgroups 0 and 1 as well as the relationships between molecules (increase or decrease in groups 0 and 1.

#### 3.3.4. Biomarker Analysis

According to the biomarker analysis results, the highest AUC values for the molecules to be considered the best biomarkers of differentiation are presented in Table 6.

### 3.4. Univariate Analysis (One-Way ANOVA) to Evaluate the Relationship between Urine Metabolites and DKD/2DM Progression (from P1 to P3)

#### 3.4.1. Analysis of Variance (ANOVA) Parameters and Fisher’s LSD, VIP Scores and MDA Values

Table 7 represents the f-values and *p*-values as well as the Fisher’s LSD significant correlations between the controls (group C) and subgroups P1–P3.

#### 3.4.2. PLSDA and Heatmap

The discrimination between the control group (C) and subgroups P1–P3 is presented in Figure 6a, according to PLSDA analysis. The heatmap presented in Figure 6b shows the significant molecules with increased or decreased levels when comparing group C with subgroups P1–P3.

### 3.5. Variations of Significant Serum and Urine Metabolites at Different DKD Stages

Figure 7 represents a selection of the most significant blood serum molecules that may be used as markers of DKD progression and considers the evolution in the P1 vs. P3 subgroup of samples.

Figure 8 represents a selection of the most significant urine molecules that may be used as markers of DKD progression, considering the evolution in the P1 vs. P3 subgroup of samples.

## 4. Discussion

The emergence of metabolomics provides insight into the pathogenic mechanisms involved in the initiation and progression of DKD in a noninvasive manner. Novel targeted metabolomics strategies can identify plasma and urine biomarkers to elucidate the pathogenic mechanisms of DKD and establish a clinical prediction model [24].

The purpose of the study was to identify new metabolomic biomarkers involved in early DKD, paying special attention to amino acids metabolism. The metabolites found in our study (tryptophan, kynurenic acid, taurine, and tiglylglycine) allowed for discrimination between healthy controls and DKD patients, as well as between all stages of DKD, which were classified according to UACR and eGFR.

As documented by many studies, one of the dreaded complications of DM is diabetic nephropathy. Even with an increasing number of metabolites, most studies are inconsistent in their findings. Of note, 44 studies described 98 metabolite profiles, of which 17 metabolites with major importance were identified using meta-analysis strategies. The numerous metabolomic analyses confirmed the implication of several pathways in DN pathogenesis, e.g., urea cycle, TCA cycle, glycolysis, and amino acid metabolisms. Hippuric acid, allantoin (in urine), and glutamine (in blood) were considered putative biomarkers for early diagnosis by performing a meta-analysis which included recent studies [25].

Novel biomarkers are needed to predict the development of the disease. However, even though numerous studies have highlighted some potential biomarkers, the prediction of disease prognosis and progression remains difficult. Recently, a rapid decline in kidney function was detected in patients with DKD whose estimated glomerular filtration rate was between 30 and 60 mL/min/1.73 m^2^. The decline was detected using non-targeted metabolomics analysis on the patients’ urine and blood samples. Conventional logistic analysis suggested that one metabolite, urinary 1-methylpyridin-1-ium (NMP), was a potential biomarker. This research suggested that the machine learning method can detect potential biomarkers better than conventional statistics [26].

The discovery of metabolomics-based biomarkers has been centered on kidney damage research and the role of amino acid metabolism. In a recent study, it was shown that 28 metabolites correlated strongly with CKD and eGFR, including 12 amino acids and 4 biogenic amines, and excluding 4 acylcarnitines. The most relevant were citrulline, kynurenine, and phenylalanine, including the kynurenine:tryptophan ratio, even eight years after the initial metabolite assessment [27].

Current diagnostic methods are not sensitive enough to detect the initial stages of the diabetic nephropathy of type 2 DM. Metabolomics is a promising tool with which to reveal the metabolic changes and the underlying mechanism involved in the pathogenesis of diabetic complications, including nephropathy [28,29]. Sixty-one serum metabolites and forty-six urine metabolites were identified as potential biomarkers related to diabetic nephropathy involved in nine serum pathways and twelve urine pathways, with significant differences in serum and urine metabolism, respectively [30].

Using the online software MetaboAnalyst 5.0, the metabolic pathways related to these metabolites were reported [31], and potential metabolites were identified for the monitoring of diabetic nephropathy, e.g., serum citrate, creatinine, arginine and its derivatives, plasma histidine, methionine and arginine, and urine oxide-3-hydroxyisovalerate, citrate, and hydroxypropionate derivatives, respectively.

Based on MDA analysis, we report significantly decreased levels of serum TRP in subgroup P1 vs. healthy controls and a slightly increased level in subgroup P1 vs. subgroup P3. MDA analysis identified a higher level of urine TRP in subgroup P1 vs. the level in the control group and subgroups P2 and P3, respectively. The latter observation was associated with the decline in eGFR across all subgroups studied.

Furthermore, KYN acid (a metabolite of TRP) had the same behavior as TRP. KYN acid had a decreased level in serum in subgroup P1 vs. control and a high value in subgroup P1 vs. P2 and P3 and a higher level in urine in P1 vs. controls, as well as a high level in P1 vs. subgroups P2–P3, respectively.

A recent review summarized the three major tryptophan metabolic pathways (kynurenine, serotonin, and indole pathways) and the connection of tryptophan metabolites with the pathogenic mechanism of patients with DKD versus non-diabetic CKD [32].

In a cross-sectional study, damaged amino acid metabolism was the principal cause of the development of diabetes. N-acetylaspartic acid, l-valine, isoleucine, asparagine, betaine, and l-methionine could make a difference between patients with DKD and those with T2DM. Moreover, l-valine and isoleucine were associated with a decline in eGFR [33].

*Cysteine* and *methionine*, as well as the *taurine–hypotaurine* metabolism pathways, were also involved in DKD compared to diabetes controls without kidney disease. Specific circulating peptides (Asn-Met-Cys-Ser and Asn-Cys-Pro-Pro) were implicated in DKD with different stages of proteinuria, in parallel with the progression of DKD proteinuria [34]. Plasma levels of *histidine* and *valine* were also identified as the main amino acids that can distinguish patients with DKD [35].

The interactions between *phenylalanine* and *tyrosine* and their secondary interaction with renal dysfunction was investigated; moreover, their low levels worked synergistically to increase the risk of T2D, and the renal dysfunction further amplified the risk [36].

In another study, blood and urine metabolites were found to be specific for DKD and included *tiglylglycine* and 3-hydroxy isovaleric acid, homovanillic acid 3-methyl crotonyl glycine [14,20].

According to Van Der et al. [37], the best discriminating urine metabolites in DM patients mainly included acylcarnitines, acylglycines, and metabolites related to tryptophan (TRP) metabolism. In other studies, a higher kynurenine/tryptophan ratio in patients with macroalbuminuria was compared to normoalbuminuric patients with type 2 DM [38]. Chou et al. discovered that a lower level of TRP was related to a rapid decline in eGFR [39].

Methoxytryptophan was found to be an important biomarker according to our data, which agrees with the findings published by Chen et al. [15].

Taurine is an important endogenous metabolite that has been reported to be significantly affected in patients with CKD [40,41]. For example, a large study highlighted alterations in the taurine level in patients with early-stage CKD [42,43].

Another study identified taurine deficiency as an important biomarker that can distinguish patients without diabetes and CKD from patients with stage 1 to stage 4 CKD [44]. In type 2 DM, a decreased level of plasma taurine was found in patients with DKD compared with healthy subjects [45].

The findings of our study revealed a decreased level of taurine in plasma in normoalbuminuric type 2 DM patients compared with healthy controls but a higher level of taurine in the P1 subgroup compared to the P3 subgroup, the results of which are consistent with the data from the literature.

Since taurine deficiency is associated with dysfunction in various tissues [46,47], a decrease in taurine levels in diabetic patients may be involved in diabetic complications, including DKD.

Tiglylglycine is an amino acid derivative that is less studied in the literature. Several findings have shown a higher tiglylglycine clearance among patients suspected of having DKD compared with those with vascular diseases [48].

Sharma et al. highlighted that some metabolites, including tiglylglycine, may have significant variations in CKD and DKD [20].

We highlight the presence of this metabolite in our samples by increasing serum concentrations in subgroups P1 to P3.

### Impact of the Metabolites Studied at Glomerular and Tubular Level in Early DKD of Type 2 DM Patients

Several key biomarkers have been identified in the urine, which reflect kidney injury at specific sites along the nephron, including glomerular injury and tubular damage, oxidative stress, inflammation, and the activation of the intrarenal renin–angiotensin system [49].

In the literature, we found data about the relationship between the significant metabolites discovered in our study and biomarkers of inflammation, tubular dysfunction, and endothelial and podocyte damage. TRP and kynurenic acid have an impact on the endothelium and an important role in inflammation.

Many studies highlighted the correlation between TRP and kynurenic acid and biomarkers of inflammation, such as tumoral necrosis factor (TNF) and Interleukin 6 (IL-6) [38], as well as of endothelial dysfunction, such as inter-cellular adhesion molecule-1 (ICAM) and vascular cell adhesion molecule-1 (VCAM) [50]. Regarding acetyl carnitine, a correlation with tubular damage markers such as KIM-1 has been reported [51]. Moreover, Ito T. et al. found that taurine suppressed the expression of VCAM-1 and ICAM-1 [45], and the literature is lacking data concerning tiglylglycine and its impact on different segments of the nephron in DKD.

Our study has several limitations. First, this is a cross-sectional study which does not allow for the establishment of a relation of causality between the metabolites found and their impact on renal structures and function. Second, hyperglycemia-induced metabolic variations could interfere with the interpretation of data. Third, residual confounders related to the dietary noncompliance of the patients might introduce a bias to the statistical analysis. Forth, it would be interesting to study the association between visceral fat mass and urinary amino acids because visceral fat and muscle mass loss could have a significant impact on the development of nephropathy [52].

However, our study has several strengths. We found the most significant metabolites and the classes they belong to, such as taurine (free amino acids), tiglylglycine (amino acid metabolites), tryptophan and kynurenic acid, and methoxytryptophan. Moreover, as a novelty, the study allowed for the identification of metabolites which expressed specific variations in normoalbuminuric type 2 DM patients, thus increasing the accuracy of the diagnosis of early DKD.

## 5. Conclusions

This study performed an integrated metabolomic profiling of blood and urine in patients with T2DM, classified in three subgroups according to albuminuria (normo-, micro-, and macroalbuminuria) and healthy controls through UPLC-MS. The multivariate and univariate algorithms confirmed the relevance of some amino acids and derivatives as biomarkers responsible for the discrimination between healthy controls and DKD patients. Serum molecules such as tiglylglycine, methoxytryptophan, serotonin sulfate, 5-hydroxy lysine, taurine, kynurenic acid, and tyrosine were found to be more significant in discriminating between group C and subgroups P1–P2–P3. In urine, o-phosphothreonine, aspartic acid, 5-hydroxy lysine, uric acid, and methoxytryptophan were among the most relevant metabolites for discriminating between group C and the DKD group, as well as between subgroups P1–P2–P3. The identification of these potential biomarkers may indicate their involvement in early DKD and 2DM progression, reflecting kidney injury at specific sites along the nephron, even in the early stages of DKD. Moreover, this study provides a particular metabolomic profile related to metabolites which could impact both the glomeruli and the tubules, even in the early stages of DKD. Further longitudinal studies of targeted metabolomics are required to validate the findings of our research and the diagnostic value of the metabolites studied in the detection of early renal involvement in the course of type 2 DM, with a special focus on the specific metabolic pathways of amino acids which may impact renal structures along all segments of the nephron. 

## Figures and Tables

**Figure 1 biomedicines-11-01527-f001:**
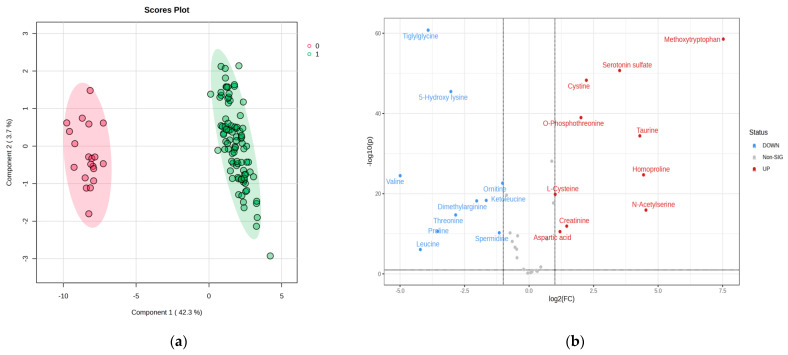
Presentsthe discrimination between the control group (code 0) and the pathologic group P (code 1) groups (**a**) according to PLSDA analysis. The fold change (FC) values and the log2FC are presented in (**b**) (volcano plot), showing the significant molecules with increased or decreased levels when compared group C with P.

**Figure 2 biomedicines-11-01527-f002:**
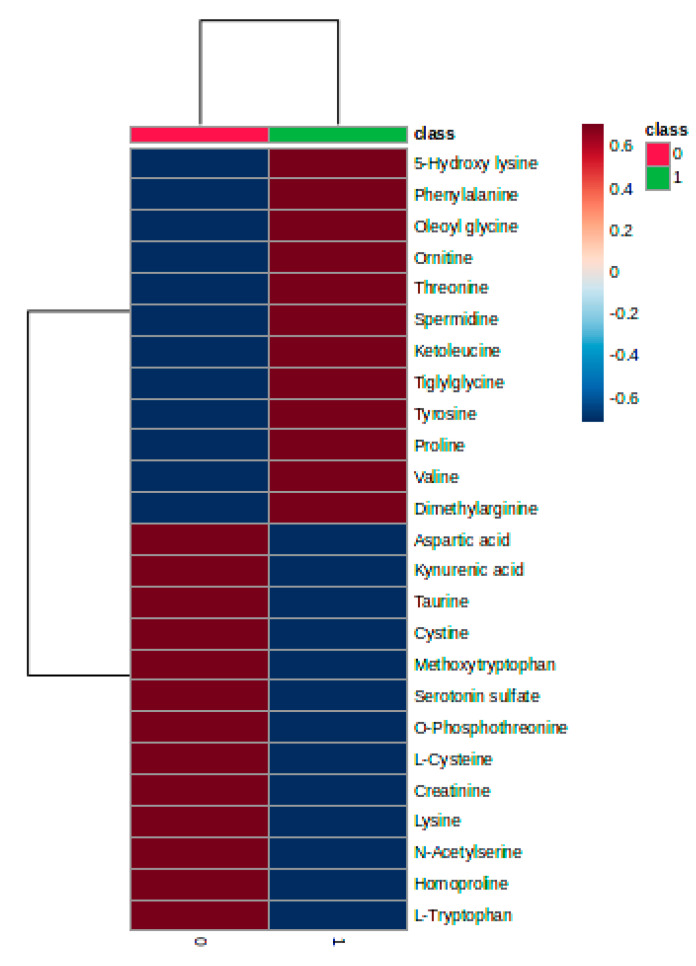
Heatmap graphic showing the clusters of blood group samples (upper side) and molecules which have decreased increased levels between the groups 0 and 1.

**Figure 3 biomedicines-11-01527-f003:**
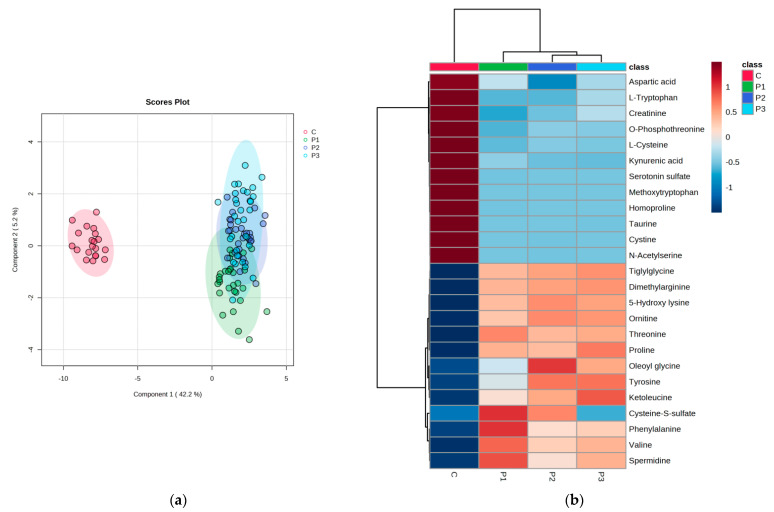
(**a**)showsthe discrimination between the control group (C) and the subgroups P1–P3. The heatmap (**b**) illustrates the most significant molecules with increased or decreased levels when comparing group C with subgroups P1–P3.

**Figure 4 biomedicines-11-01527-f004:**
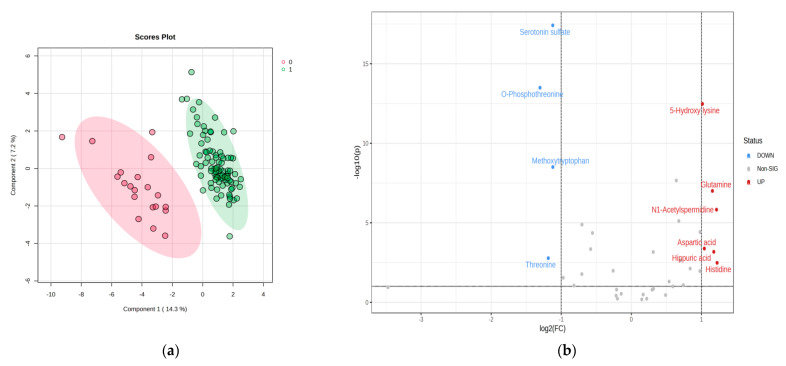
Discrimination between the control group (code 0) and the pathologic group P (code 1) groups according to PLSDA analysis (**a**). The fold change (FC) values and the log2FC are presented in (**b**) (volcano plot), showing the significant molecules with increased or decreased levels when comparing group C with P.

**Figure 5 biomedicines-11-01527-f005:**
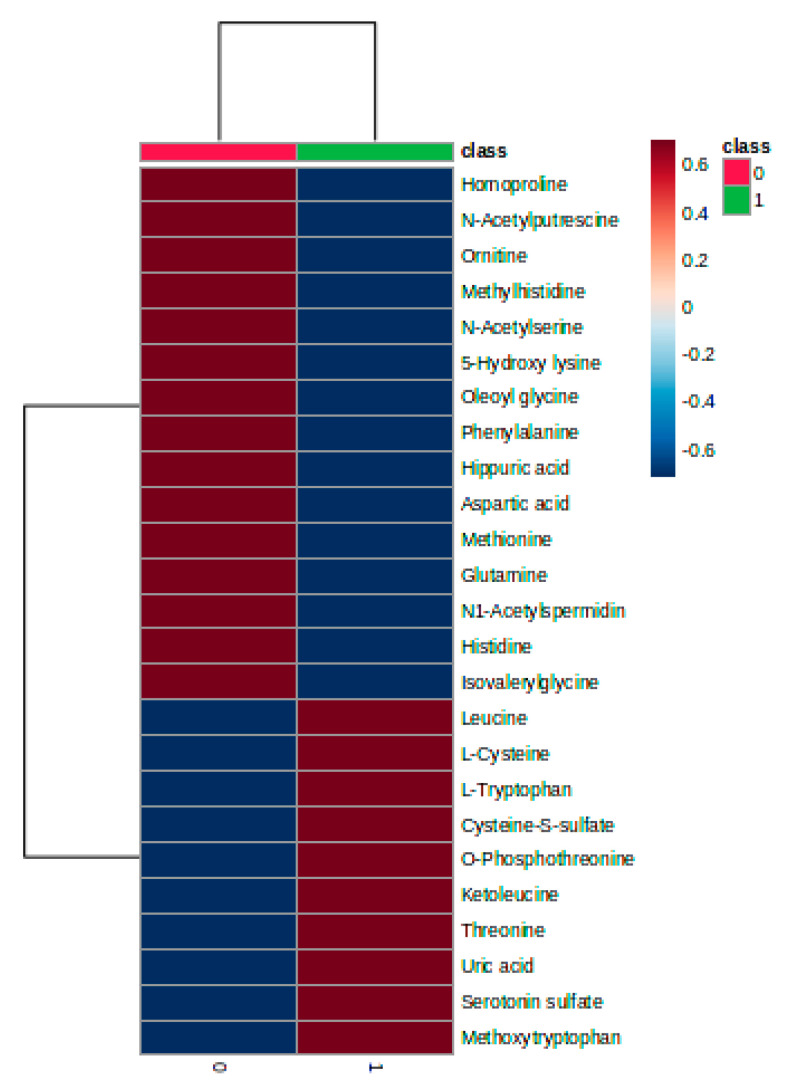
Heatmap graphic illustrating the clusters of urine group samples (upper side) and molecules which have decreased or increased levels between groups 0 and 1.

**Figure 6 biomedicines-11-01527-f006:**
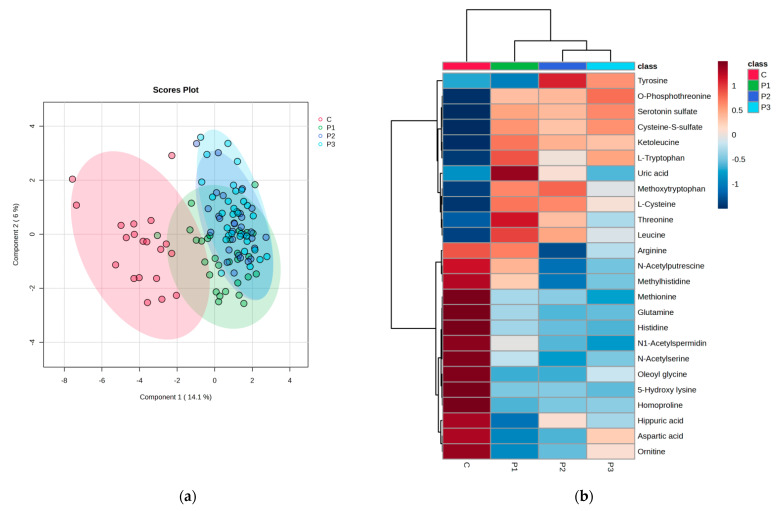
(**a**,**b**) The cross-validation algorithm presented in this case has an accuracy of 0.65 with R2 values (0.65) and a significant Q2 value (0.5) for the second component, demonstrating the acceptable predictability of the model. Moreover, in this case, the heatmap confirms the correct classification and clusters of group C vs. subgroups P1–P3 and their relative variation (increase or decrease).

**Figure 7 biomedicines-11-01527-f007:**
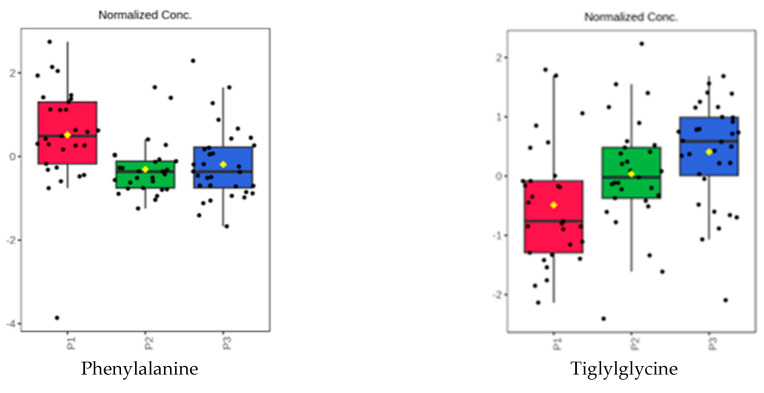
Representation of the normalized levels of some serum molecules in subgroups P1, P2, and P3, which were selected using analysis of variance (ANOVA) analysis (see details in Table 6).

**Figure 8 biomedicines-11-01527-f008:**
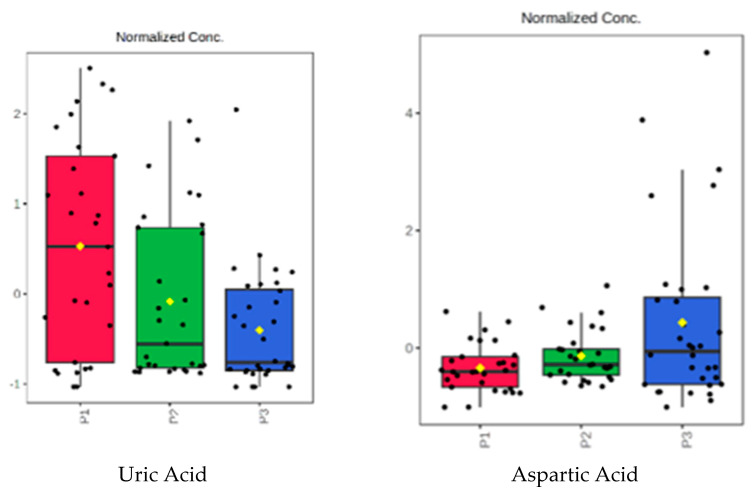
Representation of the normalized levels of some urine molecules in subgroups P1, P2, and P3, which were selected using ANOVA analysis (see details in Table 7).

**Table 1 biomedicines-11-01527-t001:** Demographic and clinical data of the groups and subgroups of patients.

	P1	P2	P3	C
Number of participants	30	30	30	20
Age (y)	68.41 ± 4.98	68.65 ± 4.91	68.84 ± 4.98	55.85 ± 7.25
DM duration (y)	9.6 ± 3.99	9.7 ± 3.99	12.78 ± 3.35	0
Serum creatinine (mg/dL)	0.82 ± 0.18	0.93 ± 0.21	1.07 ± 0.32	0.73 ± 0.08
eGFR (mL/min/1.73 m^2^)	90.42 ± 18.10	89.70 ± 18.19	77.85 ± 19.38	97.93 ± 11.71
UACR (mg/g)	7.38 ± 3.22	45.42 ± 57.08	319.86 ± 585.80	5 ± 0.23
HbA1c (%)	5 ± 0.23	6.42 ± 1.29	7.15 ± 1.60	4.98 ± 0.23
Body weight (kg)	79.06 ± 4.90	84.86 ± 3.9	86.06 ± 4.46	78.05 ± 5.47
Male/female (number)	17/13	16/14	20/10	12/8

DM duration—diabetes mellitus duration; eGFR—estimated glomerular filtration rate; HbA1c—hemoglobin A1c; UACR—urinary albumin/creatinine ratio; data reported as means ± standard deviation.

**Table 2 biomedicines-11-01527-t002:** Fold change (FC), log 2(FC) values, VIP scores according to PLSDA analysis, and MDA values according to Random Forest analysis. The significance (increase I or decrease D in group P vs. C) is also mentioned in column 4. All these molecules had VIP scores over 0.9 (see Table 2).

Molecules	Fold Change (FC)	log2(FC)	Significance P vs. C	VIP Scores	MDA
*Methoxytryptophan*	*184.460*	*7.527*	*D*	*1.492*	*0.02*
*N-Acetylserine*	*23.102*	*4.530*	*D*	*1.072*	*0.0002*
*Homoproline*	*21.654*	*4.437*	*D*	*1.245*	*0.002*
*Taurine*	*19.588*	*4.292*	*D*	*1.360*	*0.02*
*Serotonin sulfate*	*11.395*	*3.510*	*D*	*1.464*	*0.02*
*Cystine*	*4.663*	*2.221*	*D*	*1.453*	*0.03*
*O-Phosphothreonine*	*4.030*	*2.011*	*D*	*1.398*	*0.01*
*L-Tryptophan*	*3.030*	*1.011*	*D*	*1.114*	*0.001*
*Creatinine*	*2.750*	*1.460*	*D*	*0.955*	*0.002*
*Aspartic acid*	*2.294*	*1.198*	*D*	*0.907*	*5.41 × 10^−5^*
*L-Cysteine*	*2.018*	*1.013*	*D*	*1.160*	*0.0007*
*Kynurenic acid*	*0.675*	*0.87*	*D*	*1.292*	*0.02*
Ornitine	0.490	−1.028	I	1.212	0.001
Spermidine	0.449	−1.155	I	0.896	0.001
Ketoleucine	0.316	−1.660	I	1.129	0.002
Dimethylarginine	0.245	−2.028	I	1.126	0.003
Threonine	0.139	−2.845	I	1.040	0.0003
5-Hydroxy lysine	0.122	−3.030	I	1.439	0.01
Proline	0.086	−3.544	I	0.910	0.02
Tiglylglycine	0.066	−3.911	I	1.499	0.02
Leucine	0.054	−4.213	I	0.989	0.002
Tyrosine	0.039	−2.811	I	1.156	0.009
Valine	0.031	−4.995	I	1.242	0.04

These parameters and the sign of the log2(FC) show that 12 molecules, from *methoxytryptophan* to *kynurenic acid*, are relevant and decrease (D) in the P group compared to the C group, while the molecules from ornithine to valine increase (I) in the P group compared to the C group.

**Table 3 biomedicines-11-01527-t003:** The AUC values and their significance (*p* < 0.0001) for the blood molecules to be considered significant biomarkers of differentiation between the controls (code 0) and the DKD group (code 1).

Molecule	AUC	*p*-Value	Molecule	AUC	*p*-Value
*Proline*	*1*	*6.59 × 10^−12^*	Ornitine	0.985	1.18 × 10^25^
*Valine*	*1*	*1.50 × 10^−24^*	Ketoleucine	0.978	3.71 × 10*^−^*^26^
Taurine	1	2.13 *×* 10*^−^*^35^	Phenylalanine	0.977	2.59 × 10*^−^*^9^
*Tiglylglycine*	*1*	*2.22 × 10^−73^*	Spermidine	0.964	3.77 × 10*^−^*^11^
*Tyrosine*	*1*	*1.65 × 10^−23^*	Lysine	0.954	5.87 × 10*^−^*^10^
Kynurenic acid	1	1.25 *×* 10*^−^*^27^	Dimethylarginine	0.951	3.71 × 10*^−^*^18^
Methoxytryptophan	1	1.61 *×* 10*^−^*^64^	Creatinine	0.938	9.96 × 10*^−^*^13^
Cystine	1	1.07 *×* 10*^−^*^49^	Uric acid	0.937	2.84 × 10*^−^*^8^
Serotonin sulfate	1	5.23 *×* 10*^−^*^51^	Aspartic acid	0.923	4.54 × 10*^−^*^11^
L-Tryptophan	0.993	3.72 *×* 10*^−^*^19^	Homoproline	0.918	7.90 × 10*^−^*^26^
5-Hydroxy lysine	0.992	9.44 *×* 10*^−^*^49^	Threonine	0.9	1.30 × 10*^−^*^14^
O-Phosphothreonine	0.989	4.41 *×* 10*^−^*^43^			

Considering the previous findings, the biomarker analysis confirms that, out of 9 molecules with maximal AUC scores (equal to 1), 5 molecules decrease in the P group (Taurine, Kynurenic acid, Methoxytryptophan, Cystine, Serotonin sulfate). Meanwhile, 4 molecules (Proline, Valine, Tyrosine, Tiglylglycine) increased in group P compared to the controls.

**Table 4 biomedicines-11-01527-t004:** The f-values and *p*-values, VIP scores, and MDA values the Fisher’s LSD correlations for the molecules that are significantly different between the controls (group C) and subgroups P1–P3.

Molecule	f-Value	*p*-Value	VIP Score	MDA	Fisher’s LSD
Tiglylglycine	486.67	8.31 *×* 10^−62^	1.605	0.04	P1–C; P2–C; P3–C; P2–P1; P3–P1
Methoxytryptophan	373.98	3.26 *×* 10^−56^	1.448	0.05	C–P1; C–P2; C–P3
Serotonin sulfate	257.95	1.51 *×* 10^−48^	1.417	0.01	C–P1; C–P2; C–P3
Cystine	229.12	3.60 *×* 10^−46^	1.416	0.01	C–P1; C–P2; C–P3
5-Hydroxy lysine	220.71	1.99 *×* 10^−45^	1.490	0.01	P1–C; P2–C; P3–C; P2–P1
O-Phosphothreonine	149.86	5.52 *×* 10^−38^	1.257	0.01	C–P1; C–P2; C–P3
Taurine	111.25	1.29 *×* 10^−32^	1.333	0.01	C–P1; C–P2; C–P3
Kynurenic acid	78.191	9.39 *×* 10^−27^	1.342	0.01	C–P1; C–P2; C–P3
Tyrosine	75.927	2.72 *×* 10^−26^	1.584	0.04	P1–C; P2–C; P3–C; P2–P1; P3–P1
Homoproline	61.627	3.85 *×* 10^−23^	1.206	0.0003	C–P1; C–P2; C–P3

Molecules such as tiglylglycine, methoxytryptophan, serotonin sulfate, cystine, 5-hydroxy lysine, taurine, kynurenic acid, tyrosine, and valine are among the most relevant regarding the discrimination between group C and subgroups P1–P2–P3.

**Table 5 biomedicines-11-01527-t005:** Fold change (FC), log 2(FC) values, VIP scores according to PLSDA analysis, and MDA values according to random forest analysis. The significance (increase I or decrease D in group P vs. C) is also mentioned in column 4.

Molecule	Fold Change	log2(FC)	Significance P vs. C	VIP	MDA
*Histidine*	*2.335*	*1.224*	*D*	*0.849*	*0.0006*
*N1-Acetylspermidine*	*2.322*	*1.215*	*D*	*1.345*	*0.002*
*Hippuric acid*	*2.261*	*1.177*	*D*	*0.977*	*0.0005*
*Glutamine*	*2.230*	*1.157*	*D*	*1.472*	*0.004*
*Aspartic acid*	*2.059*	*1.042*	*D*	*1.011*	*0.0005*
*5-Hydroxy lysine*	*2.020*	*1.014*	*D*	*1.906*	*0.03*
*Oleoyl glycine*	*1.005*	*0.924*	*D*	*1.537*	*0.03*
Methoxytryptophan	0.460	−1.120	I	1.613	0.008
Serotonin sulfate	0.460	−1.121	I	2.170	0.05
Threonine	0.440	−1.185	I	0.905	0.006
O-Phosphothreonine	0.405	−1.303	I	1.968	0.03
L-Tryptophan	0.360	−1.070	I	1.005	0.0003
Dimethylarginine	0.090	−3.472	I	0.464	0.001

These parameters and the sign of the log2(FC) shows that 12 molecules, from *istidine* to *oleoyl glycine*, are relevant and decrease (D) in the P group compared to the C group, while the molecules from methoxytryptophan to dimethylarginine increase (I) in the P group compared to the C group.

**Table 6 biomedicines-11-01527-t006:** The AUC values (down to 0.667) and their significance (*p* < 0.3) for the blood molecules to be considered significant biomarkers of differentiation between the controls (code 0) and the DKD group (code 1).

Name	AUC	*p*-Value	Name	AUC	*p*-Value
Serotonin sulfate	0.990	2.34 *×* 10^−20^	L-Tryptophan	0.778	9.30 *×* 10^−5^
O-Phosphothreonine	0.963	1.33 *×* 10^−15^	N1-Acetylspermidine	0.756	2.92 *×* 10^−5^
Methoxytryptophan	0.933	9.15 *×* 10^−12^	Homoproline	0.739	1.49 *×* 10^−4^
Cysteine-S-sulfate	0.912	8.55 *×* 10^−6^	Hippuric acid	0.728	0.0016
Leucine	0.911	0.036	Isovalerylglycine	0.704	0.3170
Oleoyl glycine	0.889	5.31 *×* 10^−6^	L-Cysteine	0.685	3.48 *×* 10^−5^
Ketoleucine	0.859	1.31 *×* 10^−6^	N-Acetylserine	0.683	0.0238
5-Hydroxy lysine	0.839	1.04 *×* 10^−12^	Aspartic acid	0.675	0.0032
Threonine	0.780	0.0021	Dimethylarginine	0.667	0.1142

Considering the previous findings, the biomarker analysis confirms that all 5 molecules have AUC scores above 0.900, are increased in group P compared to the controls and may be considered predictive biomarkers.

**Table 7 biomedicines-11-01527-t007:** The f-values and *p*-values, VIP scores and MDA values, and the Fisher’s LSD correlations for the molecules are significantly different between the controls (group C) and subgroups P1–P3.

	f-Value	*p*-Value	VIP Score	MDA	Fisher’s LSD
Serotonin sulfate	37.141	1.74 *×* 10^−16^	2.243	0.03	P1-C; P2-C; P3-C
O-Phosphothreonine	27.236	3.88 *×* 10^−13^	2.269	0.03	P1-C; P2-C; P3-C
5-Hydroxy lysine	22.646	2.09 *×* 10^−11^	1.927	0.02	C-P1; C-P2; C-P3
Methoxytryptophan	18.767	7.75 *×* 10^−10^	0.987	0.01	P1-C; P2-C; P3-C; P1-P3; P2-P3
Oleoyl glycine	13.243	2.08 *×* 10^−7^	1.141	0.03	C-P1; C-P2; C-P3
Glutamine	10.961	2.50 *×* 10^−6^	1.767	0.002	C-P1; C-P2; C-P3
N1-Acetylspermidine	10.160	6.14 *×* 10^−6^	1.767	0.002	C-P1; C-P2; C-P3
Methionine	7.638	0.0001	1.573	0.003	C-P1; C-P2; C-P3
Uric acid	7.587	0.0001	1.403	0.003	P1-C; P1-P2; P1-P3

According to *p*-values, the most significant differences were noticed between group 0 (C) and subgroups P1–P3. Molecules such as serotonin sulfate, o-phosphothreonine, 5-hydroxy lysine, cystine, taurine, kynurenic acid, methoxytryptophan, and oleoyl glycine are among the most relevant in the discrimination between group C and subgroups P1–P2–P3.

## Data Availability

The data that support the findings of this study are available from the corresponding author upon reasonable request.

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
