# Peer review of "Metabolomic Investigation of Blood and Urinary Amino Acids and Derivatives in Patients with Type 2 Diabetes Mellitus and Early Diabetic Kidney Disease"

_biomedicines, 2023, doi:10.3390/biomedicines11061527_

Round 1
Reviewer 1 Report
The paper "Metabolomic Investigation of blood and Urinary aminoacids and derivatives in patients with type 2 diabetes mellitus and early diabetic Kidney disease" presents a cross sectional study focus on aminoacid metabolism and searching of new biomarkers in early DKD of type 2 Diabetes.
The paper is well organize and use different statistical tools for finding biomarkers. However there are a few comments/suggestions/questions that i would like to address:
In serum sample preparation. The authors didn´t use any anticoagulant , and neither use any centrifuge. COuld you explain a bit more how you get the serum samples?
Regarding Urine samples. How the authors normalize the Urine samples?. Each individual urine is different in terms of concentration ( some people produce urine more diluted than others).
In Results section, in Biomarker analysis. The authors write "according to Biomarker Analysis".... But I could´t find in the paper hoe the biomarker analysis was done. Could you add it to the paper? and the same with the AUC value.
Author Response
The paper "Metabolomic Investigation of blood and Urinary aminoacids and derivatives in patients with type 2 diabetes mellitus and early diabetic Kidney disease" presents a cross sectional study focus on aminoacid metabolism and searching of new biomarkers in early DKD of type 2 Diabetes.
The paper is well organized and uses different statistical tools for finding biomarkers. However, there are a few comments/suggestions/questions that I would like to address:
Thank you!
- In serum sample preparation. The authors didn´t use any anticoagulant, and neither use any centrifuge. COuld you explain a bit more how you get the serum samples?
Blood was collected by venipuncture in sterile vacutainers without anticoagulant, and the serum was kept at - 80 ÌŠ C until analysis. The vacutainers were labeled using confidential numerical codes. Urine samples were collected in the morning in sterile vials. A volume of 0.8 mL mix of pure HPLC-grade Methanol and Acetonitrile (2:1 v/v) was added for each volume of 0.2 mL of serum and 0.2mL urine, respectively. In each case the mixture was vortexed to precipitate proteins, ultrasonicated 5 min, and kept 24 hours at -20˚˚̊C for increasing the protein precipitation. The supernatant was collected after the centrifugation at 12,500 rpm for 10 min (4°C) and filtered through Nylon filters (0.2 μm).
- Regarding Urine samples. How the authors normalize the Urine samples? Each individual urine is different in terms of concentration (some people produce urine more diluted than others).
We normalized urine samples to urinary creatinine.
- In Results section, in Biomarker analysis. The authors write "according to Biomarker Analysis".... But I could´t find in the paper how the biomarker analysis was done. Could you add it to the paper? and the same with the AUC value.
The Biomarker analysis was done according to the standardized Metaboanalyst 5.0 software (link https://www.metaboanalyst.ca/MetaboAnalyst/upload/RocUploadView.xhtml) based on the metrices which contained the peak intensities for each metabolite. The ROC curves and the AUC values were calculated by this standardized procedure. We added this information in the manuscript at section 2.4.

Reviewer 2 Report
I think this is a very interesting study and well described.
However, Table 1 contains too little information. What about other microvascular complications? What about gender? What about oral medications, e.g., SGLT2 inhibitors, which are factors that affect urinary albumin?
Regarding p-values, it is not so important how many non-zero digits after the decimal point the p-value appears for the first time; it would be easier to see if it were stated uniformly as P<0.001, etc.
Figure 3, 4 and 6 are too small and lacks visibility.
Urinary protein is strongly influenced by body weight in addition to glycemic control; it would be useful to specify the subject's weight in Table 1 and to examine whether the composition of urinary amino acids changes between obese and non-obese subjects.
The association between visceral fat mass (e.g., abdominal circumference) and urinary amino acids is interesting because visceral fat and muscle mass loss each have a significant impact on the development of nephropathy, as shown in Diabet Med. 2020 Jan;37(1):105-113. If you have abdominal circumference or other assessed visceral fat levels, please describe them.
none
Author Response
I think this is a very interesting study and well described.
- However, Table 1 contains too little information. What about other microvascular complications? What about gender? What about oral medications, e.g., SGLT2 inhibitors, which are factors that affect urinary albumin?
- Thank you! None of the patients received SGLT2 inhibitors. All patients were on oral medication with ARBs or ACEs.
- As regardin microvascular complications 2 patients from normoalbuminuric group had diabetic polyneuropathy and 20 from the groups micro-macroalbuminuria had retinopathy.
|
|
P1 |
P2 |
P3 |
C |
|
Number of participants |
30 |
30 |
30 |
20 |
|
Age (y) |
68.41 ± 4.98 |
68.65 ± 4.91 |
68.84 ± 4.98 |
55.85 ± 7.25 |
|
DM duration (y) |
9.6 ± 3.99 |
9.7 ± 3.99 |
12.78 ± 3.35 |
0 |
|
Serum creatinine (mg/dL) |
0.82 ± 0.18 |
0.93 ± 0.21 |
1.07 ± 0.32 |
0.73 ± 0.08 |
|
eGFR (mL/min/1.73 m2) |
90.42 ± 18.10 |
89.70 ± 18.19 |
77.85 ± 19.38 |
97.93 ± 11.71 |
|
UACR (mg/g) |
7.38 ± 3.22 |
45.42 ± 57.08 |
319.86 ± 585.80 |
5 ± 0.23 |
|
HbA1c (%) |
5 ± 0.23 |
6.42 ± 1.29 |
7.15 ± 1.60 |
4.98 ± 0.23 |
|
Body weight (kg/m2) |
79.06 ±4.90 |
84.86 ±3.9 |
86.06 ±4.46 |
78.05 ±5.47 |
|
Male/female (nr |
17/13 |
16/14 |
20/10 |
12/8 |
- Regarding p-values, it is not so important how many non-zero digits after the decimal point the p-value appears for the first time; it would be easier to see if it were stated uniformly as P<0.001, etc.
I corrected in the main text your suggestion.
- Figure 3, 4 and 6 are too small and lacks visibility.
- Urinary protein is strongly influenced by body weight in addition to glycemic control; it would be useful to specify the subject's weight in Table 1 and to examine whether the composition of urinary amino acids changes between obese and non-obese subjects.
Unfortunately, in our study we did not have data about the composition of urinary amino acids changes between obese and non-obese subjects.
- The association between visceral fat mass (e.g., abdominal circumference) and urinary amino acids is interesting because visceral fat and muscle mass loss each have a significant impact on the development of nephropathy, as shown in Diabet Med. 2020 Jan;37(1):105-113. If you have abdominal circumference or other assessed visceral fat levels, please describe them.
It would be interesting to be taken into account the influence between visceral fat mass and urinary aminoacids.

Reviewer 3 Report
The paper describes the metabolomics analysis for differentiation between diabetes mellites and control subjects.
As the metabolomics method, LC/HRMS (with TOF type MS) /MS was performed.
As the statistical analysis, both multivariate and univariate analyses were performed.
The altered metabolites are valuable for scientific soundness and as clinical practice.
I have only a few comments.
1) You don't need to write the first letter of a compound in capital.
2) Was the threshold of VIP score set in Table 2?
3) Is the graphical abstract or scheme in this disease able to be drawn?
Author Response
The paper describes the metabolomics analysis for differentiation between diabetes mellites and control subjects.
As the metabolomics method, LC/HRMS (with TOF type MS) /MS was performed.
As the statistical analysis, both multivariate and univariate analyses were performed.
The altered metabolites are valuable for scientific soundness and as clinical practice.
I have only a few comments.
Thank you!
1) You don't need to write the first letter of a compound in capital. Thank-you for this remark, we made changes as you suggested.
2) Was the threshold of VIP score set in Table 2?
The threshold of VIP scores of the molecules presented in Table 2 was >0.900, as can be seen in Table 1.
3) Is the graphical abstract or scheme in this disease able to be drawn?
I changed the graphic abstract and the scheme of this disease is represented in the image below.
